# XLVIN: eXecuted Latent Value Iteration Nets

## Abstract

Value Iteration Networks (VINs) have emerged as a popular method to perform implicit planning within deep reinforcement learning, enabling performance improvements on tasks requiring long-range reasoning and understanding of environment dynamics. This came with several limitations, however: the model is not explicitly incentivised to perform meaningful planning computations, the underlying state space is assumed to be discrete, and the Markov decision process (MDP) is assumed fixed and known. We propose eXecuted Latent Value Iteration Networks (XLVINs), which combine recent developments across contrastive self-supervised learning, graph representation learning and neural algorithmic reasoning to alleviate *all* of the above limitations, successfully deploying VIN-style models on generic environments. XLVINs match the performance of VIN-like models when the underlying MDP is discrete, fixed and known, and provide significant improvements to model-free baselines across three general MDP setups.

## 1 Introduction

Planning is an important aspect of reinforcement learning (RL) algorithms, and planning algorithms are usually characterised by explicit modelling of the environment. Recently, several approaches explore *implicit planning* (also called *model-free planning*) (Tamar et al., 2016; Oh et al., 2017; Racanière et al., 2017; Silver et al., 2017; Niu et al., 2018; Guez et al., 2018; 2019). Instead of training explicit environment models and leveraging planning algorithms, such approaches propose inductive biases in the policy function to enable planning to emerge, while training the policy in a model-free manner. A notable example of this line of research are *value iteration networks* (VINs), which observe that the value iteration (VI) algorithm on a grid-world can be understood as a convolution of state values and transition probabilities followed by max-pooling, which inspired the use of a CNN-based VI module (Tamar et al., 2016). *Generalized value iteration networks* (GVINs), based on graph kernels (Yanardag & Vishwanathan, 2015), lift the assumption that the environment is a grid-world and allow planning on irregular discrete state spaces (Niu et al., 2018), such as graphs.

While such models *can* learn to perform VI, they are in no way constrained or explicitly incentivised to do so. Policies including such planning modules might exploit their capacity for different purposes, potentially finding ways to overfit the training data instead of learning how to perform planning. Further, both VINs and GVINs assume discrete state spaces, incurring loss of information for problems with naturally continuous state spaces. Finally, and most importantly, both approaches require the graph specifying the underlying Markov decision process (MDP) to be known in advance and are inapplicable if it is too large to be stored in memory, or in other ways inaccessible.

In this paper, we propose the **eXecuted Latent Value Iteration Network** (XLVIN), an implicit planning policy network which embodies the computation of VIN-like models while addressing all of the above issues. As a result, we are able to seamlessly run XLVINs with minimal configuration changes on discrete-action environments from MDPs with known structure (such as grid-worlds), through pixel-based ones (such as Atari), all the way towards fully continuous-state environments, consistently outperforming or matching baseline models which lack XLVIN's inductive biases.

To achieve this, we have unified recent concepts from several areas of representation learning:

- Using *contrastive self-supervised representation learning*, we are able to meaningfully infer dynamics of the MDP, even when it is not provided. In particular, we leverage the work of Kipf et al. (2020); van der Pol et al. (2020), which uses the TransE model (Bordes et al.,

2013) to embed states and actions into vector spaces, in such a way that the effect of action embeddings onto the state embeddings are consistent with the true environment dynamics.

- By applying recent advances in *graph representation learning* (Battaglia et al., 2018; Bronstein et al., 2017; Hamilton et al., 2017), we designed a message passing architecture (Gilmer et al., 2017) which can traverse our partially-inferred MDP, without imposing strong structural constraints (i.e., our model is not restricted to grid-worlds).

- We better align our planning module with VI by leveraging recent advances from *neural algorithm execution*, which has shown that GNNs can learn polynomial-time graph algorithms (Xu et al., 2019; Veličković et al., 2019; Yan et al., 2020; Georgiev & Lió, 2020; Veličković et al., 2020), by supervising them to structure their problem solving process according to a target algorithm. Relevantly, it was shown that GNNs are capable of executing value iteration in supervised learning settings (Deac et al., 2020). To the best of our knowledge, our work represents the first implicit planning architecture powered by concepts from neural algorithmic reasoning, expanding the application space of VIN-like models.

While we focus our discussion on VINs and GVINs, which we directly generalise and with which we share key design concepts (like the VI-based differentiable planning module), there are other lines of research that our approach could be linked to. Significant work has been done on representation learning in RL (Givan et al., 2003; Ferns et al., 2004; 2011; Jaderberg et al., 2017; Ha & Schmidhuber, 2018b; Gelada et al., 2019), often exploiting observed state similarities.

Regarding work in planning in latent spaces (Oh et al., 2017; Farquhar et al., 2018; Hafner et al., 2019; van der Pol et al., 2020), *Value Prediction Networks* (Oh et al., 2017) and *TreeQN* (Farquhar et al., 2018) explore some ideas similar to our work, with important differences; they use explicit planning algorithms, while XLVINs do fully implicit planning in the latent space. However, due to the way in which value estimates are represented, the policy network is capable of melding both model-free and model-based cues robustly. Furthermore, while our VI executor provides a representation that aligns with the predictive needs of value iteration, it can also incorporate additional information if it benefits the performance of the model.

## 2 BACKGROUND

**Value iteration** Value iteration is a successive approximation method for finding the optimal value function of a discounted *Markov decision processes* (MDPs) as the fixed-point of the so-called Bellman optimality operator (Puterman, 2014). A discounted MDP is a tuple $(\mathcal{S}, \mathcal{A}, R, P, \gamma)$ where $s \in \mathcal{S}$ are *states*, $a \in \mathcal{A}$ are *actions*, $R : \mathcal{S} \times \mathcal{A} \to \mathbb{R}$ is a reward function, $P : \mathcal{S} \times \mathcal{A} \to \text{Dist}(\mathcal{S})$ is a *transition function* such that $P(s'|s, a)$ is the conditional probability of transitioning to state $s'$ when the agent executes action $a$ in state $s$, and $\gamma \in [0, 1]$ is a discount factor which trades off between the relevance of immediate and future rewards. In the infinite horizon discounted setting, an agent sequentially chooses actions according to a stationary Markov *policy* $\pi : \mathcal{S} \times \mathcal{A} \to [0, 1]$ such that $\pi(a|s)$ is a conditional probability distribution over actions given a state. The *return* is defined as $G_t = \sum_{k=0}^{\infty} \gamma^k R(a_{t+k}, s_{t+k})$. Value functions $V^\pi(s, a) = \mathbb{E}_\pi[G_t|s_t = s]$ and $Q^\pi(s, a) = \mathbb{E}_\pi[G_t|s_t = s, a_t = a]$ represent the expected return induced by a policy in an MDP when conditioned on a state or state-action pair respectively. In the infinite horizon discounted setting, we know that there exists an optimal stationary Markov policy $\pi^*$ such that for any policy $\pi$ it holds that $V^{\pi^*}(s) \geq V^\pi(s)$ for all $s \in \mathcal{S}$. Furthermore, such optimal policy can be deterministic – *greedy* – with respect to the optimal values. Therefore, in order to find a $\pi^*$ it suffices to find the unique optimal value function $V^\star$ as the fixed-point of the Bellman optimality operator. Value iteration is in fact the instantiation of the method of successive approximation method for finding the fixed-point of a contractive operator. The optimal value function $V^\star$ is such a fixed-point and satisfies the *Bellman optimality equations* (Bellman, 1966):

$$V^\star(s) = \max_{a \in \mathcal{A}} \left( R(s, a) + \gamma \sum_{s' \in \mathcal{S}} P(s'|s, a) V^\star(s') \right) \ . \tag{1}$$

**TransE** The TransE (Bordes et al., 2013) loss for embedding objects and relations can be adapted to RL. State embeddings are obtained by an *encoder* $z : \mathcal{S} \to \mathbb{R}^k$ and the effect of an action in a given state is modelled by a translation model $T : \mathbb{R}^k \times \mathcal{A} \to \mathbb{R}^k$. Specifically, $T(z(s), a)$ is a

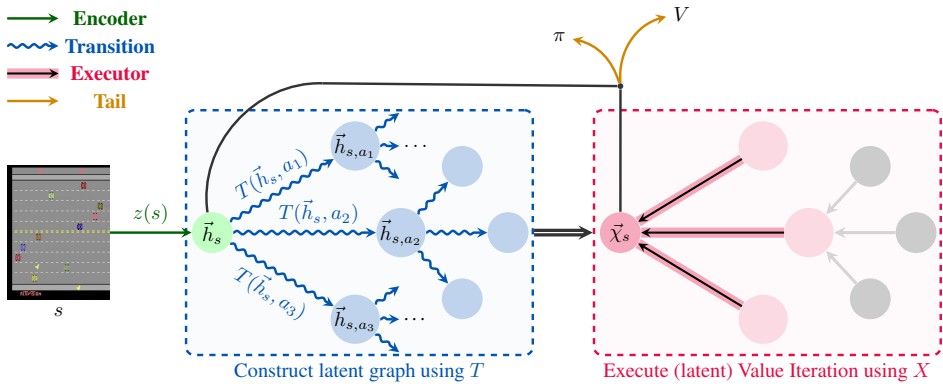

Figure 1: XLVIN model summary. The individual modules are explained (and colour-coded) in Section 3.1, and the dataflow is outlined in Section 3.2.

*translation vector* to be added to $z(s)$ in order to obtain an embedding of the resulting state when taking action $a$ in state $s$.[1] Such embeddings should also be far away from all other states $z(\tilde{s})$. Therefore, the embedding functions are optimized using the following variant of the triplet loss:

$$\mathcal{L} = d(z(s) + T(z(s), a), z(s')) + \max(0, \xi - d(z(\tilde{s}), z(s'))) \qquad (2)$$

where $\xi$ is the hyperparameter of the hinge loss, $(s, a, s')$ is a transition in a given MDP and $\tilde{s}$ is a negative sample state. Trajectories sampled from the MDP can be used as a training set of *positive* triplets $(s, a, s')$, whereas negative triplets $(s, a, \tilde{s})$ are obtained by sampling $\tilde{s}$ uniformly at random.

**GNNs and algorithmic executors**  Graph neural networks (GNNs) have been intensively studied as a tool to process graph-structured inputs, and were successfully applied to various RL tasks (Wang et al., 2018; Klissarov & Precup, 2019; Adhikari et al., 2020). GNN operations are commonly expressed through the *message passing* framework (Gilmer et al., 2017). For each node $i$ in the graph, a set of messages is computed—one message for each neighbouring node $j$, derived by applying a *message function* $M$ to the relevant node and edge features $(\vec{h}_i, \vec{h}_j, \vec{e}_{ij})$. Incoming messages in a neighbourhood $\mathcal{N}(i)$ are then aggregated through a permutation-invariant operator $\bigoplus$ (such as sum or max), obtaining a summarised message $\vec{m}_i$:

$$\vec{m}_i^{n+1} = \bigoplus\nolimits_{j \in \mathcal{N}(i)} M_n(\vec{h}_i^n, \vec{h}_j^n, \vec{e}_{ij}^n) \qquad (3)$$

The features of node $i$ are then updated through a function $U$ applied to these summarised messages:

$$\vec{h}_i^{n+1} = U_n(\vec{h}_i^n, \vec{m}_i^{n+1}) \qquad (4)$$

An important research direction explores the use of neural networks for learning to execute algorithms—which was recently extended to algorithms on graph-structured data (Veličković et al., 2019). In particular, Xu et al. (2019) establishes *algorithmic alignment* between GNNs and dynamic programming algorithms. Furthermore, Veličković et al. (2019) show that supervising the GNN on the algorithm's intermediate results is highly beneficial for out-of-distribution generalization. As VI is, in fact, a dynamic programming algorithm, a GNN executor is a suitable choice for learning it, and good results on executing VI have emerged on synthetic graphs (Deac et al., 2020).

## 3 XLVIN ARCHITECTURE

With all of the above components in place, we can fully specify the computations of the eXecuted Latent Value Iteration Network (XLVIN). Throughout this section, we recommend referring to Figure 1 for a visualisation of the model dataflow (more compact overview in Appendix A).

We propose a *policy network*—a function, $\pi_\theta(a|s)$, which for a given state $s \in \mathcal{S}$ specifies a probability distribution of performing each of the actions $a \in \mathcal{A}$ in that state. Here, $\theta$ are the neural network parameters, to be optimised using gradient ascent.

---

[1] For stochastic MDPs, the resulting embedding should be understood as an *expectation* under $P(s'|s, a)$.

### 3.1 XLVIN MODULES

**Encoder**   The encoder function, $z : \mathcal{S} \to \mathbb{R}^k$, consumes state representations $s \in \mathcal{S}$ and produces flat embeddings, $\vec{h}_s = z(s) \in \mathbb{R}^k$. The design of this component is flexible and may be dependent on the structure present in states. For example, pixel-based environments will necessitate CNN encoders, while environments with flat observations are likely to be amenable to MLP encoders.

**Transition**   The transition function, $T : \mathbb{R}^k \times \mathcal{A} \to \mathbb{R}^k$, models the effects of taking actions, in the *latent* space. Accordingly, it consumes a state embedding $z(s)$ and an action $a$ and produces the appropriate *translation* of the state embedding, to match the embedding of the successor state (in expectation). That is, it is desirable that $T$ satisfies the following property:

$$z(s) + T(z(s), a) \approx \mathbb{E}_{s' \sim P(s'|s,a)} z(s') \tag{5}$$

As $T$ operates in a flat embedding space, it is commonly realised as an MLP.

**Executor**   The executor function, $X : \mathbb{R}^k \times \mathbb{R}^{|\mathcal{A}| \times k} \to \mathbb{R}^k$, generalises the planning modules proposed in VIN (Tamar et al., 2016) and GVIN (Niu et al., 2018). It processes an embedding $\vec{h}_s$ of a given state $s$, alongside a neighbourhood set $\mathcal{N}(\vec{h}_s)$, which contains (expected) embeddings of states that immediately neighbour $s$—for example, through taking actions. Hence,

$$\mathcal{N}(\vec{h}_s) \approx \left\{ \mathbb{E}_{s' \sim P(s'|s,a)} z(s') \right\}_{a \in \mathcal{A}} \tag{6}$$

The executor combines the neighbourhood set features to produce an updated embedding of state $s$, $\vec{\chi}_s = X(\vec{h}_s, \mathcal{N}(\vec{h}_s))$, which is mindful of the properties and structure of the neighbourhood. Ideally, $X$ would perform operations in the latent space which mimic the one-step behaviour of VI, allowing for the model to meaningfully plan from state $s$ by stacking several layers of $X$ (with $K$ layers allowing for exploring length-$K$ trajectories). Given the relational structure of a state and its neighbours, the executor is commonly realised as a graph neural network (GNN).

**Actor & Tail components**   The actor function, $A : \mathbb{R}^k \times \mathbb{R}^k \to [0, 1]^{|\mathcal{A}|}$ consumes the state embedding $\vec{h}_s$ and the updated state embedding $\vec{\chi}_s$, producing action probabilities $\pi_\theta(a|s) = A\left(\vec{h}_s, \vec{\chi}_s\right)_a$, specifying the policy to be followed by our XLVIN agent. Lastly, note that we may also have additional *tail* networks which have the same input as $A$. We train XLVINs using proximal policy optimisation (PPO) (Schulman et al., 2017), which necessitates a state-value function: $V : \mathbb{R}^k \times \mathbb{R}^k \to \mathbb{R}$. Hence, we also include $V$ as a component of our model.

### 3.2 OVERVIEW OF XLVIN

Putting all of the above components together, we now present a step-by-step algorithm used by XLVINs to derive a policy $\pi_\theta(a|s)$, for a given input state $s$ and executor depth $K$:

1. Embed the input state by using the encoder function: $\vec{h}_s = z(s)$.

2. Initialise the depth-0 set of state embeddings to the singleton set, $\mathbb{S}^0 = \{\vec{h}_s\}$, containing only the embedding of $s$. Accordingly, initialise the set of edges to $\mathbb{E} = \emptyset$.

3. For each graph depth, $k \in [0, K]$:

   (a) Initialise the depth-$(k + 1)$ embedding set to $\mathbb{S}^{k+1} = \emptyset$.

   (b) For each state embedding in the previous depth, $\vec{h} \in \mathbb{S}^k$, and action, $a \in \mathcal{A}$:

       i. Compute (expected) neighbour embedding of $\vec{h}$ upon taking action $a$ using the transition model: $\vec{h}' = \vec{h} + T(\vec{h}, a)$.

       ii. Attach $\vec{h}'$ to the graph, by $\mathbb{S}^{k+1} = \mathbb{S}^{k+1} \cup \{\vec{h}'\}$, $\mathbb{E} = \mathbb{E} \cup \{(\vec{h}, \vec{h}', a)\}$.

   (c) For each state embedding $\vec{h} \in \mathbb{S}^k$, define its neighbourhood as the set of all adjacent embeddings: $\mathcal{N}(\vec{h}) = \{\vec{h}' \mid \exists \alpha \in \mathcal{A}.(\vec{h}, \vec{h}', \alpha) \in \mathbb{E}\}$.

4. Run the execution model over the graph specified by the nodes $\mathbb{S} = \bigcup_{k=0}^{K} \mathbb{S}^k$ and edges $\mathbb{E}$, by repeatedly applying $\vec{h} = X(\vec{h}, \mathcal{N}(\vec{h}))$, for every embedding $\vec{h} \in \mathbb{S}$, for $K$ steps. Denote the updated embedding of $s$, obtained by the above procedure, as $\vec{\chi}_s$.

5. Finally, use the actor and tail to predict the policy and value functions from the state embedding and updated state embedding of $s$: $\pi_\theta(s, \cdot) = A(\vec{h}_s, \vec{\chi}_s); V(s) = V(\vec{h}_s, \vec{\chi}_s)$.

**Discussion**    The entire procedure is end-to-end differentiable, does not impose any assumptions on the structure of the underlying MDP, and has the capacity to perform computations directly aligned with value iteration. Hence our model can be considered as a generalisation of VIN-like methods to settings where the MDP is not provided or otherwise difficult to obtain.

Our tree expansion strategy is *breadth-first*, which expands every action from every node, yielding $O(|\mathcal{A}|^K)$ time and space complexity. While this is prohibitive for scaling up to larger values of $K$, we have empirically found that performance tends to plateau by $K \leq 4$ for all environments we have considered, mirroring prior work on implicit planning (Farquhar et al., 2018).

We defer allowing for deeper expansions and large action spaces to future work, but note that it will likely require a *rollout policy*, selecting actions to expand from a given state. For example, I2A (Racanière et al., 2017) obtains a rollout policy by distilling the agent's policy network. Extensions of our model to continuous actions could also be achieved by rollout policies, although discretising the action space through binning (Tang & Agrawal, 2020) is also a viable route.

### 3.3   XLVIN Training

As discussed in Section 3.1, our transition function, $T$, and executor function, $X$, should ideally respect certain properties (e.g. Equations 5–6). We *pre-train* both of them using established methods in the literature; (Kipf et al., 2020; van der Pol et al., 2020) for TransE and (Deac et al., 2020) for the executor. Due to space constraints, we report detailed pre-training information in Appendix B.

To optimise the neural network parameters $\theta$, we use proximal policy optimisation (PPO)[2] (Schulman et al., 2017). Note that the PPO gradients also flow into parameters of $T$ and $X$, which has the potential to displace them from the properties required by the above, leading to either poorly constructed graphs (that don't respect the underlying MDP) or lack of VI-aligned computation.

For the transition model, we resolve this by periodically re-training on the TransE loss during training (with a multiplicative factor of $0.001$ to the loss in Equation 2). As we have no such easy way of retraining the executor, $X$, without knowledge of the underlying MDP, we instead opt to *freeze* the parameters of $X$ after pre-training, treating them as constants rather than parameters to be optimised.

While we found it sufficient to fine-tune on TransE using only the trajectories obtained from our policy network, we note that, especially once the policy becomes more exploitative, the data collected in this way may strongly bias the transition model and make it unusable for exploration. Hence, we anticipate that some environments will require a careful tradeoff between exploration and exploitation for the data collection strategy for training the transition model.

## 4   Experiments

We now proceed to evaluating the proposed XLVIN architecture across three distinct kinds of environments, and a variety of training and testing modes. Our work is centered around the following three research questions, which we will repeatedly refer to throughout this section:

**Q1** How do XLVINs compare to (G)VINs when they are applicable (i.e. on fixed, known and discrete MDPs)? Is the graph built within XLVINs as useful as the underlying MDP?

**Q2** Do XLVINs effectively **generalise** the VIN model? Do they outperform baseline architectures (without the transition and executor models) across *general* environments?

**Q3** Does the XLVINs' environment model help them robustly plan? Are XLVINs amenable to dynamically changing environments (e.g. continual learning) or low-data regimes?

---

[2]We use the publicly available PyTorch PPO implementation and hyperparameters from the following repository by Ilya Kostrikov: `https://github.com/ikostrikov/pytorch-a2c-ppo-acktr-gail`.

## 4.1 Experimental setup

We will now categorise our environments into three types—for further details, see Appendix C.

**Known MDP** In order to compare XLVINs performance against VINs and GVINs (**Q1**) we evaluate on an established environment with a known, fixed and discrete MDP. We use the $8 \times 8$ grid-world mazes proposed by Tamar et al. (2016). The observation for this environment consists of the maze image, the starting position of the agent and the goal position. Every maze is associated with a *difficulty* level, equal to the length of the shortest path between the start and the goal.

We utilise this difficulty to formulate the *continual maze* task: the agent is, initially, trained to solve only mazes of difficulty 1. Once the agent reaches 95% success rate on the last 1,000 sampled episodes of difficulty $d$, it automatically advances to difficulty $d + 1$ (without observing difficulty $d$ again). If the agent fails to reach 95% within 1,000,000 trajectories, it is considered to have *failed*. At each passed difficulty, the agent is evaluated by computing its success rate on held-out test mazes.

Setting up the task in this way allowed us to evaluate the resilience of the agents to *catastrophic forgetting* (**Q3**)—as the difficulties get harder, the models get further inspired to memorise the training set (see Appendix D for further statistics) We also test *out-of-distribution generalisation* (training on $8 \times 8$ mazes, testing on $16 \times 16$), allowing for another way to test the agents' robustness (**Q3**).

Given the grid-world structure, our encoder for the maze environment is a three-layer CNN computing 128 latent features and 10 outputs—which we describe in Appendix E. It is important to note that the VIN model of Tamar et al. (2016) performs the aggregation by *slicing* directly the features on the agent's coordinates. This assumes upfront knowledge of where the agent actually is on the map (rather than having to infer it from data) and hence puts VINs in a privileged position. For a possibly fairer comparison (**Q1**), we also attempted to train **VIN-mean** and **VIN-max**—VIN models where the slicing is replaced by global average pooling or global max pooling.

The transition function is a three-layer MLP with layer normalisation (Ba et al., 2016) after the second layer, for all environments. For mazes, it computes 128 hidden features and applies ReLU.

The executor is, for all environments, identical to the message passing executor of Deac et al. (2020). For mazes, we exploit the fact that the MDP is known: we train the executor exactly on the graph structures generated from the training mazes; please see Appendix F for further details. We apply the executor until depth $K = 4$, with layer normalisation (Ba et al., 2016) applied after every step.

**Continuous-space observations** The latent-space execution of XLVINs allow us to deploy it in generic environments—especially, continuous-space environments are now supported without having to discretise them. We investigate the performance of XLVIN across these environments (**Q2**), focusing on three classical control environments from the OpenAI Gym: CartPole-v0, Acrobot-v1 and MountainCar-v0. We also address **Q3** by presenting extremely-limited-data scenarios.

The encoder function is now a three-layer MLP with ReLU activations, computing 50 output features and $F$ hidden features, where $F = 64$ for CartPole, $F = 32$ for Acrobot, and $F = 16$ for MountainCar. The same hidden dimension is also used in the transition function, which otherwise matches the one used for mazes.

The executor has been trained from synthetic graphs which are designed to imitate the dynamics of CartPole very crudely—the MDP graphs being binary trees where only certain leaves carry zero reward and are terminal. We also attempt training the executor from completely random deterministic graphs—making no assumptions on the underlying environment. More details on the graph construction, for both of these approaches, is given in Appendix F. The same trained executor is then deployed across all environments, to demonstrate robustness to synthetic graph construction (**Q3**). In all cases, the XLVIN uses $K = 2$ executor layers.

Before moving on, it is worthy to note that CartPole offers dense and frequent rewards—making it easy for policy gradient methods. We make the task challenging by sampling *only 10 trajectories* at the beginning, and not allowing any further interaction—beyond 100 epochs of training on this dataset. Conversely, Acrobot and MountainCar are both sparse-reward, and known to be challenging for policy gradient. For these environments, we sample 5 trajectories at a time, twenty times during training (for a total of 100 trajectories).

Table 1: Mean scores for CartPole-v0, Acrobot-v1 and MountainCar-v0 after training, averaged over 100 episodes and five seeds. Baseline CartPole results reprinted from van der Pol et al. (2020).

| CartPole-v0 | 100 trajectories | Only 10 trajectories |
|---|---|---|
| REINFORCE | 23.84 ± 0.88 | - |
| WM-AE | 114.47 ± 17.32 | - |
| LD-AE | 154.73 ± 50.49 | - |
| DMDP-H ($J = 0$) | 72.81 ± 20.16 | - |
| PRAE, $J = 5$ | **171.53** ± 34.18 | - |
| PPO | - | 104.6 ± 48.5 |
| XLVIN-R | - | **199.2** ± 1.6 |
| XLVIN-CP | - | 195.2 ± 5.0 |

| Acrobot-v1 | Score |
|---|---|
| PPO | -500.0 ± 0.0 |
| XLVIN-R | -353.1 ± 120.3 |
| XLVIN-CP | **-245.4** ± 48.4 |

| MountainCar-v0 | Score |
|---|---|
| PPO | -200.0 ± 0.0 |
| XLVIN-R | -185.6 ± 8.1 |
| XLVIN-CP | **-168.9** ± 24.7 |

**Pixel-space** Lastly, we investigate how XLVINs perform on high-dimensional pixel-based observations (**Q2**), using the Atari-2600 (Bellemare et al., 2013). We focus on three games: Freeway, Alien and Enduro. These environments encompass various aspects of complexity: sparse rewards (on Freeway), larger action spaces (18 actions for Alien), visually rich observations (changing time-of-day on Enduro). Further, we successfully *re-use* (**Q3**) the executor trained on random deterministic graphs, showing additionally that that it is comparable to using the CartPole-based graphs on Freeway (whose "up-and-down" structure aligns it somewhat to such environments). We evaluate the agents' low-data performance by allowing only 1,000,000 observed transitions (**Q3**). We re-use exactly the environment and encoder from here[3], and run the executor for $K = 2$ layers for Freeway and Enduro and $K = 1$ for Alien.

### 4.2 RESULTS

In results that follow, we use **"PPO"** to denote our baseline model-free agent; it has no transition/executor model, but otherwise matches the XLVIN hyperparameters. When applicable, we use **"XLVIN-CP"** to denote XLVIN executors pretrained using CartPole-style synthetic graphs, and **"XLVIN-R"** for pre-training them on random deterministic graphs.

**Continual maze** The results of our continual maze evaluation are summarised in Figure 2. We observe that, in-distribution, the XLVIN is competitive with all other models

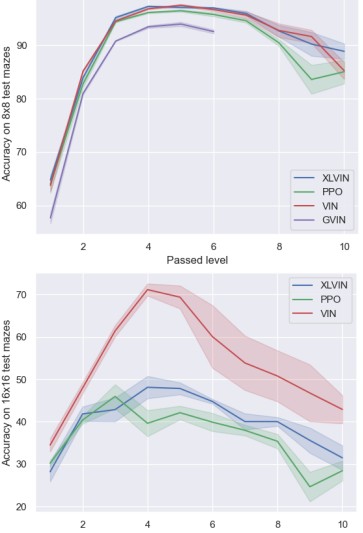

Figure 2: Success rate on $8 \times 8$ (**top**) and $16 \times 16$ (**bottom**) held-out mazes for XLVIN, PPO, VIN and GVIN, obtained after passing each level of $8 \times 8$ train mazes. VIN-mean and VIN-max **failed** to pass difficulty 1.

(including VIN and GVIN—that fails to pass difficulty 6). It is also the most resilient to level 10, which has only one maze, and is hence very easy to overfit to.

The XLVIN remains competitive with the baseline model when evaluated out-of-distribution. There is a gap to VINs, which can be attributed to the previously mentioned slicing—corroborated by the fact both VIN-mean and -max *failed* to pass even level 1 of the $8 \times 8$ mazes. Generally, the slicing operation will underperform whenever some level of global *context* needs to be taken into account when determining the grid-world's transitioning or reward rules. In Appendix G, we illustrate explicitly how XLVINs outperform VINs in a *contextual maze*: a slightly-modified version of the continual maze grid world, where such global context can modify the environment dynamics.

**CartPole, Acrobot and MountainCar** Results for the continuous-space control environments are provided in Table 1. We find that, for CartPole, the XLVIN model solves the environment from only 10 trajectories, outperforming all the results given in van der Pol et al. (2020) (incl. REINFORCE

---

[3]`https://github.com/ikostrikov/pytorch-a2c-ppo-acktr-gail/blob/master/a2c_ppo_acktr/model.py#L169-L195`

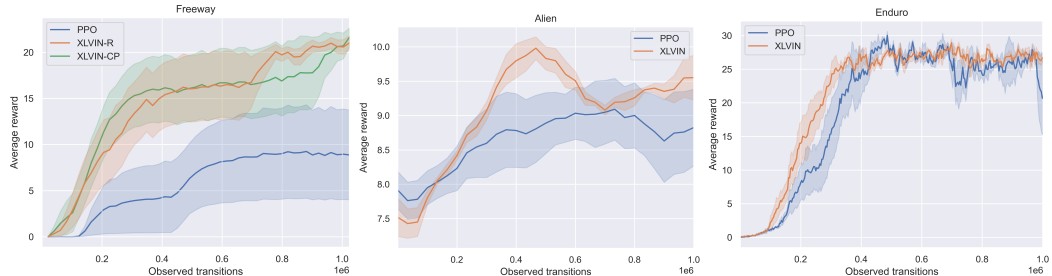

Figure 3: Average reward on Freeway, Alien and Enduro over 1,000,000 processed transitions.

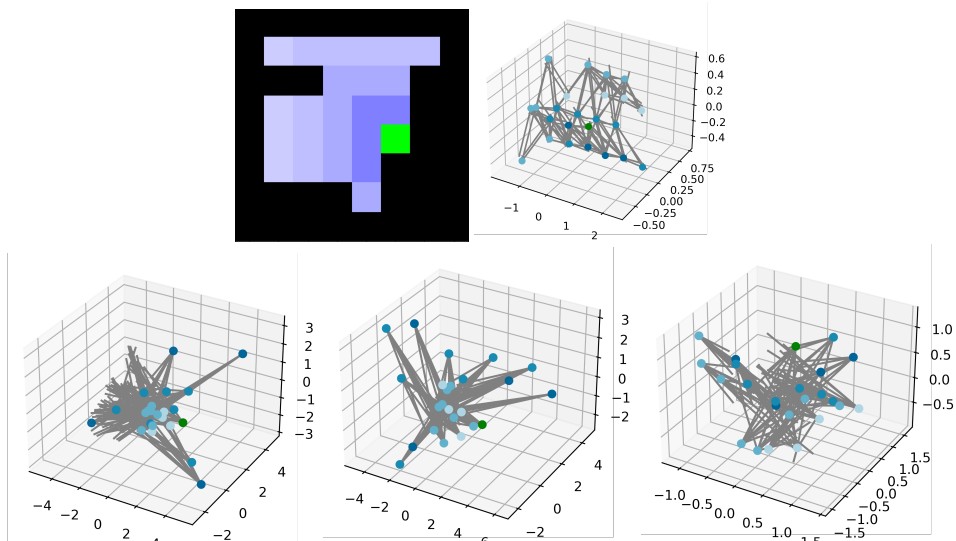

Figure 4: **Top:** A test maze (*left*) and the PCA projection of its TransE state embeddings (*right*), colour-coded by distance to goal (in green). **Bottom:** PCA projection of the XLVIN state embeddings after passing the first (*left*), second (*middle*), and ninth (*right*) level of the continual maze.

(Williams, 1992), Autoencoders, World Models (Ha & Schmidhuber, 2018a), DeepMDP (Gelada et al., 2019) and PRAE (van der Pol et al., 2020)), while using $10\times$ fewer samples.

Further, our model is capable of solving the Acrobot and MountainCar environments from only 100 trajectories, in spite of sparse rewards. Conversely, the baseline model is unable to get off the ground at all, remaining stuck at the lowest possible score in the environment until timing out. The above results still hold when XLVIN is trained on the random deterministic graphs, demonstrating that the executor training need not be dependent on knowing the underlying MDP specifics.

**Freeway, Alien and Enduro** Lastly, the average reward of the Atari agents across the first million transitions can be visualised in Figure 3. From the inception of the training, the XLVIN model explores and exploits better, consistently remaining ahead of the baseline model in the low-data regime (matching it in the latter stages of Enduro). The fact that its executor was transferred from randomly generated graphs (Appendix F) is a further statement to the model's robustness.

### 4.3  QUALITATIVE RESULTS

We qualitatively study the embeddings learnt by the encoder (and transition model), hoping to elucidate the mechanism in which XLVIN organises its plan (and hence provide further insight on **Q3**).

At the top row of Figure 4, we (*left*) colour-coded a specific $8 \times 8$ test maze by proximity to the goal state, and (*right*) visualised the 3D PCA projections of the "pure-TransE" embeddings of these states

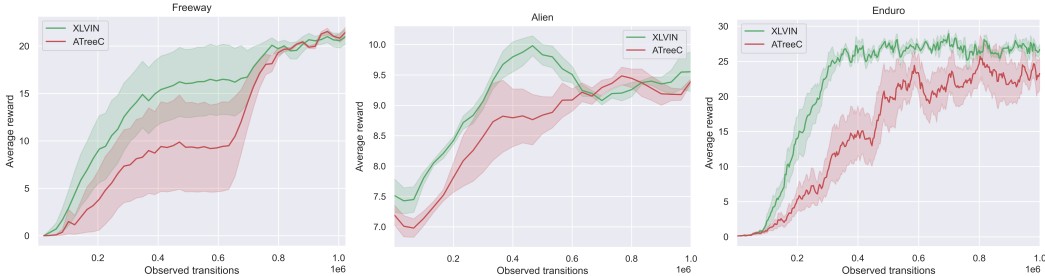

Figure 5: Comparison of XLVIN and ATreeC update rules over the first 1,000,000 Atari transitions.

(prior to any PPO training), with the edges induced by the transition model. Such a model merely seeks to organise the data, rather than optimise for returns: hence, a grid-like structure emerges.

At the bottom row, we visualise how these embeddings and transitions evolve as the agent keeps solving levels; at levels one (*left*) and two (*middle*), the embedder learnt to distinguish all 1-step and 2-step neighbours of the goal, respectively, by putting them on opposite parts of the projection space. This process does not keep going, because the agent would quickly lose capacity. Instead, by the time it passes level nine (*right*), grid structure emerges again, but now the states become partitioned by proximity: nearer neighbours of the goal are closer to the goal embedding. In a way, the XLVIN agent is learning to reorganise the grid; this time in a way that respects shortest-path proximity.

## 5 ABLATION STUDIES

Our XLVIN model is powered by a synergy of learned latent-space models and latent-space graph algorithm executors. To probe the influence of these components, we conclude with two ablation studies, directly comparing our architecture incorporating pixel-based world models, or implicit planners that rely on explicitly modelling scalar quantities, such as ATreeC (Farquhar et al., 2018).

**Comparison to pixel-based world models**   Aligned with prior studies of VAE losses on Atari (Anand et al., 2019), our attempts to use a world model based on pixel reconstruction (e.g. Ha & Schmidhuber (2018b)) for XLVINs were unsuccessful. This supports using transition models optimised in the low-dimensional latent space. For details and visualisations, see Appendix H.

**Comparison to scalar value modelling**   We compare our results with the ATreeC architecture (Farquhar et al., 2018), but still using the PPO loss. ATreeC relies on explicitly invoking a TD($\lambda$) backup to combine scalar-level predictions in every node of the expanded tree and compute the final policy logits. Since the ATreeC policy is directly tied to the result of applying TD($\lambda$), its ultimate performance is closely tied to the quality of its scalar value predictions.

The ATreeC model failed to solve any of the continuous-state environments (with its CartPole reward marginally exceeding the PPO baseline, at $117.1 \pm 56.2$, and Acrobot and MountainCar staying at -500 and -200, respectively). On all three Atari environments, it consistently trailed behind XLVIN during the first 500,000 transitions (Figure 5), and on Enduro, it underperformed even compared to the PPO baseline, indicating overreliance on scalar predictions may damage low-data performance.

## 6 CONCLUSIONS

We presented eXecuted Latent Value Iteration Networks (XLVINs), combining recent progress in self-supervised contrastive learning, graph representation learning and neural algorithm execution to generalise Value Iteration Networks to irregular, continuous or unknown MDPs. Our results have shown that XLVINs have matched or outperformed appropriate baselines, often at low-data or out-of-distribution regimes. The learnt executors are robust and transferable across environments, despite being trained on synthetic graphs that need not align with the underlying MDP. XLVINs represent, to the best of our knowledge, one of the first times neural algorithmic executors are used for implicit planning, and we believe that this is a very promising direction for future RL research.

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

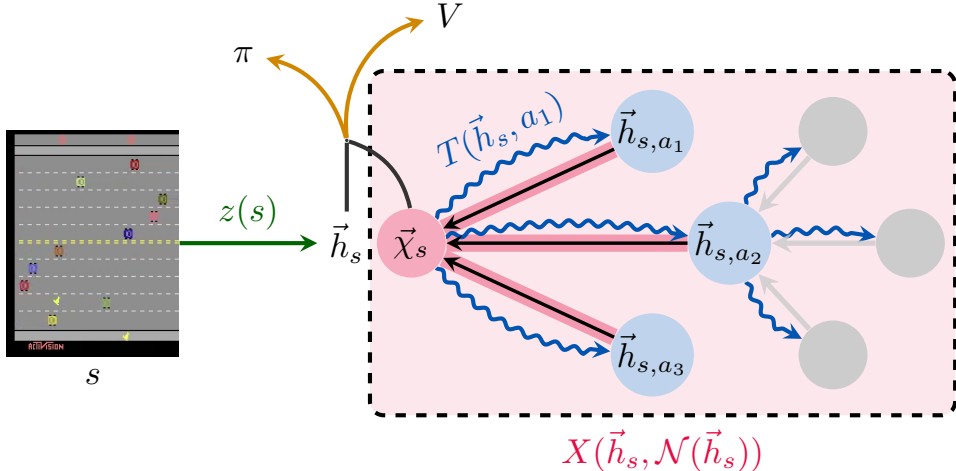

Figure 6: XLVIN model summary with compact dataflow. The individual modules are explained (and colour-coded) in Section 3.1, and the dataflow is outlined in Section 3.2.

## A  ALTERNATE RENDITION OF XLVIN DATAFLOW

See Figure 6 for an alternate visualisation of the dataflow of XLVIN—which is more compact, but does not explicitly sequentialise the operations of the transition model with the operations of the executor.

## B  PRE-TRAINING OF TRANSITION AND EXECUTOR FUNCTIONS

The transition condition of Equation 5 perfectly aligns with the TransE loss (Bordes et al., 2013), hence we pre-train $T$ by optimising the loss function of Equation 2, using transitions sampled in the environment using a random policy—not unlike the prior work by Kipf et al. (2020); van der Pol et al. (2020) that also trains transition functions in this way.

The desirable behaviour of the executor is to align with VI, and hence we pre-train $X$ to be predictive of the value updates performed by VI, following the work of Deac et al. (2020). Note that standard VI procedure requires access to a fully-specified MDP, over which we can generate VI trajectories to use as training data. When an MDP is not available, following the remarks of Deac et al. (2020), we generate synthetic discrete MDPs that align with the target MDP as much as possible[4]—finding that useful transfer still occurs, echoing the findings of Deac et al. (2020).

To summarise in brief, the executor function training proceeds as follows:

1. First, generate a dataset of MDPs which, as much as possible, mimics in some way the characteristics of the true MDP we'd like to deploy on (e.g. determinism, number of actions, etc.). If no such knowledge is available upfront, resort to a generic graph distribution like Erdős-Rényi or Barabási-Albert.

2. Then, execute the Value Iteration algorithm on these MDPs, keeping track of intermediate values $V_t(s)$ at each iteration $t$, until convergence.

3. Supervise the executor network—a graph neural network operating over the transitions of the MDP as edges—to receive $V_t(s)$—and all other parameters of the MDP—as inputs, and predict $V_{t+1}(s)$ as outputs (optimise using mean-squared error).

Figure 7: The seven environments considered within our evaluation: $8 \times 8$ and $16 \times 16$ mazes (Tamar et al., 2016) (known grid-like MDP), continuous control environments (CartPole-v0, Acrobot-v1, MountainCar-v0) and pixel-based environments (Atari Freeway, Alien and Enduro).

## C  ENVIRONMENTS UNDER STUDY

We provide a visual overview of all seven environments considered in Figure 7.

**Maze**  The maze environment with randomly generated obstacles from Tamar et al. (2016). Observations include a map of the maze (pointing out all obstacles), an indicator of the agent's location, and an indicator of the goal location. Actions correspond to moving in one of the eight principal directions, incurring a reward of $-0.01$ every move (to encourage shorter solutions), $-1$ for hitting an obstacle (which terminates the episode), and $1$ for hitting the goal (which also terminates the episode).

**CartPole**  The CartPole environment is a classic example of continuous control, first proposed by Barto et al. (1983). The goal is to keep the pole connected by an un-actuated joint to a cart in an upright position. Observations are four-dimensional vectors indicating the cart's position and velocity as well as pole's angle from vertical and pole's velocity at the tip. Actions correspond to staying still, or pushing the engine forwards or backwards. The agent receives a fixed reward of $+1$ for every timestep that the pole remains upright. The episode ends when the pole is more than 15 degrees from the vertical, the cart moves more than $2.4$ units from the center or by timing out (at 200 steps), at which point the environment is considered solved.

**Acrobot**  The Acrobot system includes two joints and two links, where the joint between the two links is actuated. Initially, the links are hanging downwards, and the goal is to swing the end of the lower link up to a given height. The environment was first proposed by Sutton (1996). The observations—specifying in full the Acrobot's configuration—constitute a six-dimensional vector, and the agent is able to swing the Acrobot using three distinct actions. The agent receives a fixed negative reward of $-1$ until either timing out (at 500 steps) or swinging the acrobot up, when the episode terminates.

**MountainCar**  The MountainCar environment is an example of a challenging, sparse-reward, continuous-space environment first proposed by Moore (1990). The objective is to make a car reach the top of the mountain, but its engine is too weak to go all the way uphill, so the agent must use gravity to their advantage by first moving in the opposite direction and gathering momentum. Observations are two-dimensional vectors indicating the car's position and velocity. Actions correspond to staying still, or pushing the engine forward or backward. The agent receives a fixed negative reward of $-1$ until either timing out (at 200 steps) or reaching the top, when the episode terminates.

**Freeway**  Freeway is a game for the Atari 2600, published by Activision in 1981, where the goal is to help the chicken cross the road (by only moving vertically upwards or downwards) while avoiding cars. It is a standard part of the Atari Learning Environment and the OpenAI Gym.

Observations in this environment are the full framebuffer of the Atari console while playing the game, which has been appropriately preprocessed as in Mnih et al. (2015). Actions correspond to staying still, moving upwards or downwards. Upon colliding with a car, the chicken will be set back a few lanes, and upon crossing a road, it will be teleported back at the other side to cross the road again (which is also the only time when it receives a positive reward of $+1$). The game automatically times out after a fixed number of transitions.

---

[4]If no prior knowledge about the environment is known, one might resort to generic graph distributions, such as Erdős-Rényi (Erdős & Rényi, 1960) or Barabási-Albert (Albert & Barabási, 2002).

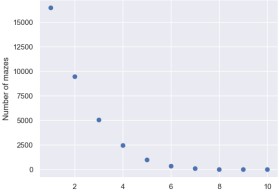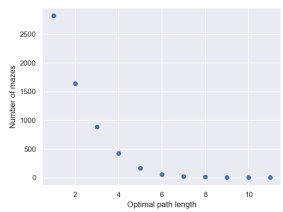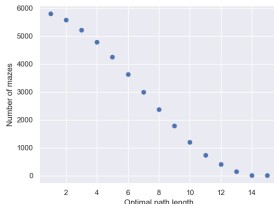

Figure 8: Number of mazes with the optimal path of given length from: $8 \times 8$ train dataset (left), $8 \times 8$ test dataset (middle) and $16 \times 16$ test dataset (right).

**Enduro** Enduro is a game for the Atari 2600, published by Activison in 1983. The goal of the game is to complete an endurance race, overtaking a certain number of cars each day of the race to continue to the next day. It is a standard part of the Atari Learning Environment and the OpenAI Gym.

Observations in this environment are the full framebuffer of the Atari console while playing the game, which has been appropriately preprocessed as in Mnih et al. (2015). This game is one of the first games with day/night cycles as well as weather changes which makes it particularly visually rich. There are nine different actions we can take in this environment corresponding to staying still as well as accelerating, decelerating, moving left/right and combinations of two of them.

**Alien** Alien is a game for the Atari 2600, published by 20th Century Fox in 1982. The goal of the game is to destroy the alien eggs laid in the hallways (similar to the pellets in Pac-Man) while running away from three aliens on the ship. It is a standard part of Atari Learning Environment and the OpenAI Gym.

Observations in this environment are the full framebuffer of the Atari console while playing the game, which has been appropriately preprocessed as in Mnih et al. (2015). There are 18 different actions we can take in this environment corresponding to staying still, firing the flamethrower and moving or firing the flamethrower in eight directions.

## D    DATA DISTRIBUTION OF MAZES

We provide an overview of simple count-based statistics of the maze datasets, stratified by difficulty, in Figure 8. Namely, we can observe that the distribution of $8 \times 8$ maze difficulties follows a power-law, whereas the $16 \times 16$ maze difficulty counts decay linearly. This poses an additional challenge when extrapolating out-of-distribution, as the distributions of the two testing datasets vary drastically—and what worked well for one's performance measure may not necessarily work well for the other.

## E    ENCODER ON CONTINUAL MAZE

Given the grid-world structure, our encoder for the maze environment is a CNN. We stack three convolutional layers computing 128 features, of kernel size $3 \times 3$, each followed by batch normalisation (Ioffe & Szegedy, 2015) and the ReLU activation. We then aggregate all positions by performing global average pooling[5] (Springenberg et al., 2014). Finally, the aggregated representation is passed to a two-layer MLP with hidden dimension 128 and output dimension 10, with a ReLU activation in-between. The output dimension was chosen to be comparable with VIN (Tamar et al., 2016).

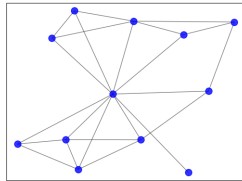 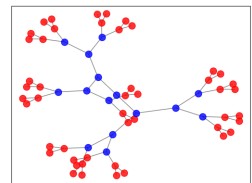 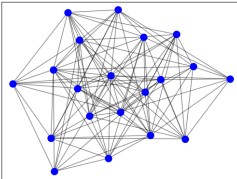

Figure 9: Synthetic graphs constructed for pre-training the GNN executor: Maze (**left**), CartPole (**middle**) and random deterministic (20 states, 8 actions) (**right**)

## F    SYNTHETIC GRAPHS

Figure 9 presents the three kinds of synthetic graphs used for pretraining the GNN executor. The one for mazes (left) emphasises the terminal node as a node to which all nodes are connected; all other nodes have a maximum of eight neighbours, corresponding to the possible action types.

In all other cases, we pre-train the executor using randomly generated deterministic graphs (right): for $|\mathcal{S}| = 20$ states and $|\mathcal{A}| = 8$ actions, we create a $|\mathcal{S}|$-node graph. For each state-action pair we select, uniformly at random, the state it transitions to, deterministically. We sample the reward model using the standard normal distribution. Overall, the graphs are sampled as follows:

$$\tilde{T}(s, a) \quad \sim \quad \text{Uniform}(|\mathcal{S}|) \tag{7}$$

$$P(s' \mid s, a) \quad = \quad \begin{cases} 1 & s' = \tilde{T}(s, a) \\ 0 & \text{otherwise} \end{cases} \tag{8}$$

$$R(s, a) \quad \sim \quad \mathcal{N}(0, 1) \tag{9}$$

These $k$-NN style graphs do not assume upfront any structural properties of the underlying MDP, and are a good prior distribution for evaluating the performance of XLVIN.

For CartPole-style environments, we attempt a third type of graph (middle). It is a binary tree, where red nodes represent nodes with reward 0, and blue nodes have reward 1. This aligns with the idea that going further from the root, which is equivalent with taking repeated left (or right) steps, leads to being more likely to fail the episode.

We also attempt using the CartPole graph for pre-training the executor for the other two continuous-observation environments (MountainCar, Acrobot) and for Freeway. Primarily, the similar action space of the environments is a possible supporting argument of the observed transferability. Moreover, MountainCar and Acrobot can be related to a inverted reward graph of CartPole, with more aggressive combinations left/right steps often bringing a higher chance of success.

## G    CONTEXTUAL MAZE ENVIRONMENT

The grid-world based VIN architecture relies on the assumption that the agent is in a fixed and known environment, and that the current agent coordinate within this environment is known. This allows value iteration to be aligned with weight-shared convolutional neural networks, followed by a *slicing* operation in the agent coordinate. While this gives the final embedding higher relevance compared to pooling all of the pixels together, it also makes the final embedding tightly *localised* around an agent's current state. Should the environment possess any global context that dictates the transition and reward dynamics, the VIN model may lack ways to easily integrate this necessary contextual information.

To illustrate this issue, we propose a slight modification to our continual maze environment: the **contextual maze**. We randomly sample two scalars, $a, b \sim \mathcal{U}(0, 1)$, and condition the reward model depending on their sum; if $a + b \leq 1$, the agent must proceed to the goal as usual; otherwise, the environment's reward model is inverted: reaching the goal now provides negative reward, and hitting (any) wall brings positive reward (and terminates the episode).

---

[5]Note that we could not have performed the usual flatten operation, in order to generalise to $16 \times 16$ mazes.

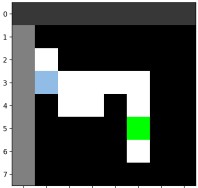 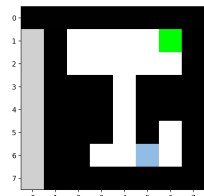 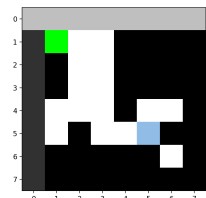 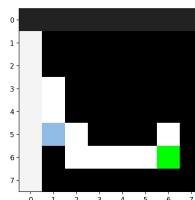

Figure 10: Procedurally generated contextual $8 \times 8$ mazes. The green cell represents the goal, and the blue cell represents the agent. Black cells represent walls. The upper and left borders' intensity is modified to mirror the sampled values of $a$ and $b$, determining the environment dynamics.

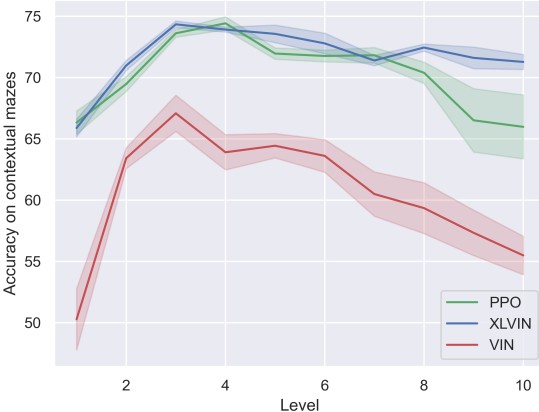

Figure 11: Success rate of the XLVIN model against VIN and PPO on the contextual maze environment, after training on each level, on held-out test mazes.

This context is provided to the agent through the colour intensity of the left and top border, to match the values of $a$ and $b$, respectively—see Figure 10 for an illustration of several procedurally generated contextual mazes. Successful agents now must meaningfully extract the contextual information *before* committing to a plan, which may be unfavourable for the VIN architecture.

Organising the environment into levels (based on the agent's initial distance to the goal), as before, and learning in a continual setting, we present the results on held-out test mazes in Figure 11. On contextual mazes, a clear gap opens up between the slicing-based VIN architecture and context-based models (XLVIN and PPO). Further, XLVINs once again demonstrate stronger resilience to catastrophic forgetting compared to the baselines, by not overfitting to the hardest levels, which also have the fewest mazes.

## H  PIXEL-BASED WORLD MODELS

We attempt replacing our Atari transition model with a variant that learns representations through pixel-based reconstructions (using a VAE objective, as done by Ha & Schmidhuber (2018b)). We found that representations obtained in this way were not useful, we observed that most of our state encodings converged to a fixed-point, and that the pixel-space reconstructions completely ignored the foreground observations (see Figure 12). This aligns with prior investigations of VAE-style losses on Atari, which found they tend to overly focus on reconstructing the background and were less predictive of RAM state than latent-space models, as well as randomly-initialised CNNs (Anand et al., 2019). This comparison stands in favour of our approach to using a transition model optimised purely in the lower-dimensional latent space.

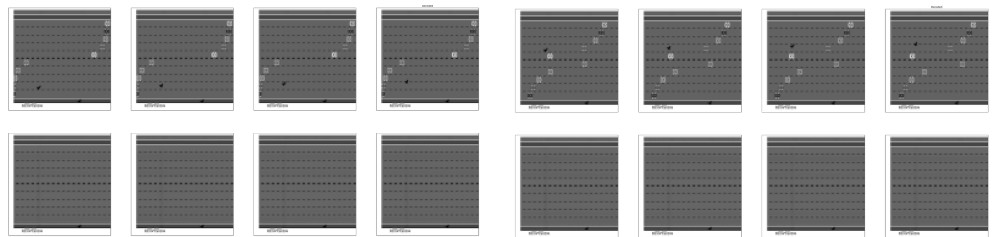

Figure 12: Freeway frames (**above**) and reconstructions (**below**) using a VAE-style world model.

