# OpenReview forum: "XLVIN: eXecuted Latent Value Iteration Nets"
_ICLR.cc/2021/Conference — Reject_

### Official Review · AnonReviewer2 · 2020-10-27
**This paper addresses the limitations of value iteration networks with regards to continuous settings and partially-known state spaces. This is achieved by approximating the full MDP with a subset of states and using graph neural networks to mimic dynamic programming. Experimental results show improvement over a model with no explicit planning module (PPO).**

**Rating:** 7
**Confidence:** 4

**Review:**

The paper tackles an open problem of the value-iteration-network-paradigm. The proposed method (XLVIN) has a conceptual edge over traditional value iteration networks in that it can be applied to continuous problems and problems where the state space is either too big or not fully known in advance. The experiments mostly succeed in making the case that XLVINs:

* are a drop-in replacement for traditional VINs in discrete settings
* are able to improve on models without explicit planning models in general

One issue with the first point is that the experiments on the $16\times16$ environments are in favor of VINs against XLVINs. The explanation provided in the paper makes sense but on the whole I get the impression that more experiments are necessary to asses whether XLVINs can truly replace VINs in discrete environments.

There is also quite a bit of algorithmic complexity that results from using the graph neural network paradigm. This is justified by the fact that XLVINs are also applicable to a much wider range of problems. However, as a reader, I'd appreciate it a lot if the paper would spend time on discussing some of that complexity in greater detail (I go into specifics under "Cons"). Regardless, I recommend accepting the paper as the ideas presented here are relevant to the VIN-community.

### Pros

* The method is applicable to continuous problems and large or not fully-known state-spaces, where VINs are not.
* The experiments on continuous problems show an edge over methods without a planning component.
* The method has an edge in the low-data regime.

### Cons

* Some important questions are not addressed:
  * How does the method scale with the planning horizon $K$?
  * How feasible is it to produce synthetic MDPs on real-world problems?
  * How feasible is it to use generic graphs instead?

  It's understandable that some content needs to be left out due to space constraints, but at the same time, these questions would be at the heart of future work on this topic and the paper would benefit from addressing them.

* The experiments on the $16\times16$ mazes no longer support the claim that XLVINs can replace VINs gracefully. An explanation is provided for this, though it's still unclear how the two models compare in discrete environments other than the $8\times 8$ mazes. I feel like more experiments are needed here, perhaps in scenarios that don't give an unfair advantage to VINs. The mean- or max-pooled VINs are not convincing for that because they fail to solve even the simplest tasks.

---

> ### Author Response · Authors · 2020-11-19
> **Reply to AnonReviewer2**
>
> Thank you for the positive feedback and your constructive review!
>
> We provide our initial responses to your comments in turn. These will be integrated into our submission on OpenReview over the next few days. Please let us know if you think we have appropriately addressed your comments, and we are of course happy to discuss further!
>
> - To illustrate that the dependence on graph type may often be alleviated in practice, we re-ran all of our continuous-environment and Freeway experiments while pre-training the executor on generic graphs, as suggested. Namely, we sample random deterministic graphs of 20 states and 8 actions (each state-action is transitioning to exactly one other state, with probability 1), with randomly sampled reward models ($r(s, a)\sim\mathcal{N}(0, 1)$). Such graphs impose no strong assumption on the structure of the MDP. The results we obtain are as follows:
>     - CartPole: 199.2 +- 1.6 (compared to 104.6 +- 48.5 for PPO, and 195.2 +- 5.0 for XLVIN with hand-crafted graphs)
>     - Acrobot: -353.1 +- 120.3 (compared to -500 for PPO and -245.4 +- 48.4 for XLVIN with hand-crafted graphs)
>     - MountainCar: -185.58 +- 8.1 (compared to -200 for PPO and -168.87 +- 24.7 for XLVIN with hand-crafted graphs)
>     - Freeway: see the figure here: https://anonymous.4open.science/repository/5dc1ee33-d391-4718-881d-5c3ad77e2b47/Freeway_scores.png (XLVIN-CP is pre-trained with hand-crafted graphs, XLVIN-R is pre-trained with random deterministic graphs).
>
> - The results indicate that, even when trained on randomly generated graph structures, the XLVIN executor is capable of solving all of the considered environments more rapidly and robustly than the PPO baseline. Hence our method need not be strongly dependent on knowing the specifics of the underlying MDP, at least on the environments evaluated here. In essence, as long as our executor learns a “correct” latent value iteration rule, it should be applicable.
>
> - With respect to the planning horizon / executor depth, K, two aspects should be taken into account:
>     - Regarding computational complexity and scaling, our current breadth-first expansion strategy requires generating $|\mathcal{A}|^K$ latent states and subsequently running a graph neural network over them. As a remark from our side: this strategy is proposed as a proof-of-concept for XLVIN-style latent computations, and we anticipate the incorporation of stronger expansion policies to be an important aspect of future work on the XLVIN agent.
>     - Further, mirroring prior observations in the space of implicit planning, we have found increasing the executor depth to gradually improve the results (either in terms of higher peak performance or lower variance), up to a threshold (which is the depth we report in our experiments) beyond which the results tend to plateau. We will include the discussion on both of these aspects in the revised paper.
>
> - Lastly, regarding the discrete environment experiments, we are currently exploring alternative grid-world variants where VIN-like slicing will not have an advantage. Ideally, resources permitting, we will survey an environment where the optimal behaviour will depend on a “global” context, rendering a sliced representation inappropriate. If you have any comments on our intent at this stage, or if we have misinterpreted your comment, please let us know.

---

> > ### Comment · AnonReviewer2 · 2020-11-24
> > **Reply to the authors**
> >
> > Thank you very much for the additional experiments, which I find quite interesting. The additional discussion is also appreciated. I maintain my position that the paper is interesting to the VIN community and should be accepted.
> >
> > The main source of doubt in my mind is still the comparison between VINs and XLVINs, which I find lacking. It is fine if there are circumstances where VINs outperform XLVINs, given that the latter is approximating the full MDP with a graph, but these should be investigated more extensively. This is my main reason for rating the paper at 7 rather than a higher score.

---

> > > ### Author Response · Authors · 2020-11-25
> > > **Second revision posted**
> > >
> > > Thank you for following up, and for your suggestion!
> > >
> > > We have just posted a second revision, where we now provide (in Appendix G, with referencing and briefly discussing in Section 4.2), experiments on a _contextual_ discrete maze environment, which illustrates a grid-world challenging for the slicing-based VIN architecture.
> > >
> > > Generally, the slicing operation will underperform whenever some level of global _context_ needs to be taken into account when determining the grid-world's transitioning or reward rules. To illustrate this effect in a simple manner, in contextual mazes we initially sample two uniform scalars, $a, b\sim\mathcal{U}(0, 1)$, and condition the reward model on their sum:
> > > - if $a + b\leq 1$, the environment behaves as before;
> > > - if $a + b > 1$, the _walls_ become the goal: hitting walls yields positive rewards, and reaching the goal state yields negative rewards.
> > >
> > > These scalar values are fed to the agent observation through the colour intensity of the left and upper borders. This poses a challenge for slicing-based models, because they have no simple way of incorporating this information before committing to a planning computation.
> > >
> > > Our experiments in the continual setting demonstrate underperformance of the slicing VIN model, with a clear gap opening up to XLVIN. Further, the XLVIN retains its beneficial properties from the original continual maze, being the most resilient to overfitting on the hardest levels.
> > >
> > > These results provide useful further discussion on the utility of VIN-style models compared to XLVIN in discrete grid-world environments (and when one might be more applicable than the other). We are, of course, happy to discuss further!

---

> ### Author Response · Authors · 2020-11-23
> **Revision posted**
>
> Thank you once again for your kind review and constructive comments!
>
> We have now revised the paper on OpenReview, incorporating all of the data mentioned in our initial reply, and including further experiments, studies and discussions.
>
> Should you have any additional comments at this stage, please let us know.

---

### Official Review · AnonReviewer1 · 2020-10-27
**Well written paper, approach proposed is simple and easy to understand**

**Rating:** 6
**Confidence:** 2

**Review:**

This paper proposed a novel policy prediction model that combines self-supervised contrastive learning, graph representation learning and neural algorithm execution to generalize the Value Iteration Networks to MDPs. The method described in the paper is a combination of existing works in the literature but seems to work well in practice. The experiments evaluate multiple aspects of the proposed model (E.g. number of executor layers, etc.) and show significant performance improvement over the existing approaches.

The latent representation used for policy prediction implicitly incorporates two-step aggregation 1) from initial representation extraction through encoding network and 2) from message passing, which seems helpful for the generalization.

As I am not actively tracking RL literature, I am not sure if there is a similar approach has already been proposed or not. My comments are based on the assumption that no existing work using GNN to do further representation learning on top of the individual encoded information. It would be better if the authors clearly state this novelty at the end of related works.

Pros
===
1. The paper is very well written and provided sufficient background knowledge to let the reader follow the description.
2. While it appears to be a simple combination of existing techniques, the proposed model shows the benefit of obtaining better latent state representations for the policy prediction task.

Cons
===
1. The novelty of the proposed model is a bit weak in terms of a lack of specialization for this particular task.
2. Figure 1 in the paper is not quite meaningful. A better demonstrative figure would help improve this paper.

---

> ### Author Response · Authors · 2020-11-19
> **Reply to AnonReviewer1**
>
> Thank you for your thoughtful review and kind words about our contributions.
>
> We provide our initial responses to your comments in turn. These will be integrated into our submission on OpenReview over the next few days. Please let us know if you think we have appropriately addressed your comments, and we are of course happy to discuss further!
>
> - The novelty of our XLVIN agent lies, as stated in our conclusion, in the fact that it is (to the best of our knowledge) the first time graph neural networks are used for implicit planning-style computations in the latent space. This is powered by concepts from self-supervised learning (for constructing graphs around the current state) and neural algorithmic execution (for simulating Value Iteration computations along these constructed graphs), which also represents a novel computational setup for RL policy networks.
> In response to your comment, we will add these remarks at the end of our related work discussion to provide more clarity.
> - Thank you for your suggestion regarding Figure 1! In response, we provide a possible updated figure; see https://anonymous.4open.science/repository/5dc1ee33-d391-4718-881d-5c3ad77e2b47/XLVIN_splitfig.png.
> This figure explicitly separates the graph construction phase and applying the GNN executor over the constructed graph, as well as including an explicit legend that maps various arrow styles to concepts introduced in the paper. Please let us know if such a figure addresses your concerns of clarity, and if there is anything else we should do to enhance its presentation.
>
> Please let us know if you feel satisfied with the proposed modifications, and if we should do anything else to improve the clarity of the paper, or otherwise. We remain very open for any further suggestions!

---

> ### Author Response · Authors · 2020-11-23
> **Revision posted**
>
> Thank you once again for your kind review and constructive comments!
>
> We have now revised the paper on OpenReview, incorporating all of the changes mentioned in our initial reply, and including further experiments, studies and discussions.
>
> Should you have any additional comments at this stage, please let us know.

---

### Official Review · AnonReviewer3 · 2020-10-28
**Extends VI to new domains, but seems to depend critically on an available representative graph**

**Rating:** 6
**Confidence:** 3

**Review:**

**Summary**

The authors propose a generalization of Value Iteration Networks to unknown, potentially continuous state spaces. They describe a framework for leveraging a learned graph embedding model (TransE) in combination with a deep RL model and an execution model based on graphical message passing to perform a VI-like operation. The authors show improved performance compared to baselines on a grid-world task with a known MDP, as well as several simple continuous control environments and the Atari game Freeway.

**General**

Overall the approach is interesting, but the current results are somewhat limited and the model, in particular the executor, seems to depend critically on the availability of a suitable graph for pre-training. The authors leverage a "CartPole-inspired" graph for the majority of the tasks, and the tasks seem to have been chosen to be consistent with the statistics of this synthetic graph (from the text, Freeway's "'up-and-down' structure aligns it somewhat to environments like CartPole"). While the authors illustrate a limited ability to generalize from graph/task misalignment on the maze task (8x8 to 16x16), it's still not clear that the model will generalize to more complex tasks where it's not straightforward to pre-train on a representative graph. For instance, all of the non-maze tasks appear to use simple ternary actions (e.g. left, right and stay for CartPole; up, down and stay for Freeway). As well, working from a task-relevant pre-training graph appears to give the agent foreknowledge of the task itself, which makes comparisons to baselines like PPO unfair.

The main question here for me is whether the paper reaches the threshold on originality and significance. The paper builds on the VIN/GVIN work and particularly depends on Deac et al. 2020 for illustrating VI with GNNs. It provides the first extension of these works to more complex tasks with unknown MDPs. However, the limited results and dependence on the availability of a suitable graph makes the significance of these extensions unclear. Nonetheless, it could lay the framework for future work that pushes the VI framework into new domains.

Section 4.3 provides a nice clarification on what the model is doing under the hood.

**Specific**

- I would like to see more exploration of the executor depth, i.e. do we see gradual improvement as the depth is increased?

- Are the Freeway with respect to the first 1 million transitions (text) or 100000 (figure)?

- \alpha is used for both the hinge loss in Eq. 2 and 3.2 3.(c) which is somewhat confusing at first glance.

**Pros:**
- Extends VI framework to more complex, unknown tasks.
- Well-written and clear, excluding the specific issues noted above.
- Nice qualitative results

**Cons:**
- Results are provided on a limited number of simple tasks, primarily using PPO as a baseline.
- Tasks seem to have been chosen such that they have a representative graph available.

---

> ### Author Response · Authors · 2020-11-19
> **Reply to AnonReviewer3**
>
> Thank you for the very careful review and kind words about our contributions.
>
> We provide our initial responses to your comments in turn. These will be integrated into our submission on OpenReview over the next few days. Please let us know if you think we have appropriately addressed your comments, and we are of course happy to discuss further!
>
> - Firstly, we strongly acknowledge your point about how even minor assumptions on the graph structure could be capable of impacting the overall downstream performance. To illustrate that the dependence on graph type may often be alleviated in practice, we re-ran all of our continuous-environment and Freeway experiments with pre-training the executor on random deterministic graphs of 20 states and 8 actions (each state-action is transitioning to exactly one other state, with probability 1), with randomly sampled reward models ($r(s, a)\sim\mathcal{N}(0, 1)$). Such graphs impose no strong assumption on the structure of the MDP. The results we obtain are as follows:
>     - CartPole: 199.2 +- 1.6 (compared to 104.6 +- 48.5 for PPO, and 195.2 +- 5.0 for XLVIN with hand-crafted graphs)
>     - Acrobot: -353.1 +- 120.3 (compared to -500 for PPO and -245.4 +- 48.4 for XLVIN with hand-crafted graphs)
>     - MountainCar: -185.58 +- 8.1 (compared to -200 for PPO and -168.87 +- 24.7 for XLVIN with hand-crafted graphs)
>     - Freeway: see the figure here: https://anonymous.4open.science/repository/5dc1ee33-d391-4718-881d-5c3ad77e2b47/Freeway_scores.png (XLVIN-CP is pre-trained with hand-crafted graphs, XLVIN-R is pre-trained with random deterministic graphs).
> - The results indicate that, even when trained on randomly generated graph structures, the XLVIN executor is capable of solving all of the considered environments more rapidly and robustly than the PPO baseline. Hence our method need not be strongly dependent on knowing the specifics of the underlying MDP, at least on the environments evaluated here. In essence, as long as our executor learns a “correct” latent value iteration rule, it should be applicable.
>
> - Further, we have performed experiments similar to our Freeway setup on two more Atari games (pre-training the executor on random deterministic graphs again):
>     - **Alien**, which is more complex both in terms of observation variety and number of actions (18). We find favourable results of XLVINs in the low-data regime here. See this figure for the results: https://anonymous.4open.science/repository/5dc1ee33-d391-4718-881d-5c3ad77e2b47/Alien_scores.png.
>     - **Enduro**, an endurance driving simulator with 9 actions. We find that in the earlier phases of the low-data regime, the XLVIN is able to take charge against the PPO baseline, and remains generally more robust in terms of variance. See this figure for the results: https://anonymous.4open.science/repository/5dc1ee33-d391-4718-881d-5c3ad77e2b47/Enduro_scores.png.
>
> - The prior work of Deac _et al._ (2020) primarily showed that it is possible to train VI executors that generalise favourably to various (known upfront) MDP structures, in a purely supervised learning setup. While this is a prerequisite for building our XLVIN agent, to the best of our knowledge, no prior work has actually successfully used this idea within reinforcement learning. We believe that with the additional results above---especially illustrating that XLVINs still outperform the PPO baseline when using a random pre-training graph distribution---our work would meet the threshold of originality. Please let us know if there are additional clarifications or discussions we should provide.
>
> - Regarding your specific comments:
>     - Generally, and mirroring prior observations in the space of implicit planning, we have found increasing the executor depth to gradually improve the results (either in terms of higher peak performance or lower variance), up to a threshold (which is the depth we report in our experiments) beyond which the results tend to plateau. We will include this discussion in the revised paper.
>     - Freeway is trained on 1,000,000 transitions (as in the text) -- there was a minor error in plotting the x-axis ticks, which is already fixed in the Freeway figure we linked above.
>     - We will replace the margin hyperparameter in the hinge loss to a different symbol than $\alpha$ -- thank you for spotting this!

---

> > ### Comment · AnonReviewer3 · 2020-11-22
> > **Reply to Authors**
> >
> > Thank you for the detailed response, particularly for the new results with random graphs and additional Atari games. I believe these will make the submission stronger and I will be revising my initial rating as a result.

---

> > > ### Author Response · Authors · 2020-11-23
> > > **Revision posted**
> > >
> > > Thank you for your reply -- we are very glad that the updates were in the right direction!
> > >
> > > We have now revised the paper on OpenReview, incorporating all of the changes mentioned above, and including further experiments, studies and discussions.
> > >
> > > Should you have any additional comments, please let us know!

---

### Official Review · AnonReviewer4 · 2020-10-29
**Interesting paper. Claims should be refined. More ablations should be included.**

**Rating:** 6
**Confidence:** 3

**Review:**

Summary:
The paper proposed a framework that combines TransE-style world model learning and value iteration networks. Section 3.2 gives a nice and clear overview of the algorithm: generating trees by calling the learned world model, run value iteration networks on the tree, and use the value iteration algorithms to compute the value and the policy at each state.
Since the output of the algorithm is a policy, the model was trained using PPO.

Comments:
The overall presentation of the algorithm is quite clear, and the proposed model is a nice combination of value iteration networks with learned world models. My comments will thus focus on the claims made by the authors and the completeness of the experiment section.

- Claims
1. The authors have made very strong arguments about the generality of the proposed method: "As a result, we are able to seamlessly run XLVINs with minimal configuration changes on environments from MDPs with known structure (such as grid-worlds), through pixel-based ones (such as Atari), all the way towards fully continuous-state environments, consistently outperforming or matching baseline models which lack XLVIN’s inductive biases." The authors have made several arguments about non-discrete and non-deterministic environments in the paper, however, they haven't shown any results on challenging continuous and stochastic environments. For example, how will the model perform in environments with high-dimensional action spaces: such as torques for robots? How will the model perform in partially-observable environments? If the proposed algorithm does not work for continuous action spaces (Page 4, footnote), the authors should make their claims weaker: finite and discrete action spaces + finite planning horizon makes the number of states visited also finite.

- Experiments
1. Overall the experiments are relatively weak. Adding more results on high-dimensional continuous environments (such as humanoids) or more difficult atari games could possibly help.
2. The contribution of the proposed algorithm comes from two parts: the world-model part and the VIN+Policy Network part. However, there are only few fine-grained ablation studies. For example, there have been a great number of models that learn the forward dynamics of the environment. How will other forward dynamics models perform in the same framework (e.g., Ha & Schmidhuber 2018a)? Does the TransE-style low-dimensional prior help?
3. Related to my previous comment: is there any way that the authors can make a quantitative evaluation of the learned world models?
4. The authors should make comparisons with other models that do explicit/implicit planning based on the same learned world models, e.g., Value Prediction Networks (Oh et al., 2017) and TreeQN (Farquhar et al., 2018).
5. There naturally exists a chicken-and-egg problem between learning world models and learning task-specific policies: learning good world models in complex environments requires good policies to collect useful data, while a better world model can improve task-specific policies. I think the authors should be more clear about how they are collecting data for training their TransE models and discuss the trade-off between exploration and exploitation in their model.

---

> ### Author Response · Authors · 2020-11-19
> **Reply to AnonReviewer4**
>
> Thank you for the detailed and constructive feedback!
>
> We provide our initial responses to your comments in turn. These will be integrated into our submission on OpenReview over the next few days. Please let us know if you think we have appropriately addressed your comments, and we are of course happy to discuss further!
>
> - We agree to tone down our claims as you suggest, especially regarding continuous action-spaces. Much of your concerns here belong to intended future work (which is complementary to our main contributions), as stated by Footnote 3. As a remark from our side: our breadth-first expansion framework is proposed as a proof-of-concept for XLVIN-style latent computations. While it is not possible to naïvely extend it to continuous/large action spaces, we anticipate that using an explicit rollout policy (which would sample the actions to expand rather than taking all of them) will be one common way of achieving so.
> - We will update our claims exactly in accordance with your review: finite and discrete action spaces are directly supported by our model as presented. We will also include a more involved discussion on potential avenues for further generalising the agent to continuous action spaces, and the challenges that may arise when doing so.
> - In order to strengthen our suite of evaluated tasks, we have now experimented on two additional Atari games (in a similar setup to Freeway):
>     - **Alien**, which is more complex both in terms of observation variety and number of actions (18). We find favourable results of XLVINs in the low-data regime here. See this figure for the results: https://anonymous.4open.science/repository/5dc1ee33-d391-4718-881d-5c3ad77e2b47/Alien_scores.png
>     - **Enduro**, an endurance driving simulator with 9 actions. We find that in the earlier phases of the low-data regime, the XLVIN is able to take charge against the PPO baseline, and remains generally more robust in terms of variance. See this figure for the results: https://anonymous.4open.science/repository/5dc1ee33-d391-4718-881d-5c3ad77e2b47/Enduro_scores.png
> - Regarding ablation studies: for now, we have performed a study on the graph types used for pre-training our executor, demonstrating that we are capable of supporting useful XLVIN computations even if the graphs are not aligned with the anticipated MDP. Namely, we re-ran all of our continuous-environment and Freeway experiments with pre-training the executor on random deterministic graphs of 20 states and 8 actions (each state-action is transitioning to exactly one other state, with probability 1), with randomly sampled reward models ($r(s, a)\sim\mathcal{N}(0, 1)$). The results we obtain are as follows:
>     - CartPole: 199.2 +- 1.6 (compared to 104.6 +- 48.5 for PPO, and 195.2 +- 5.0 for XLVIN with hand-crafted graphs)
>     - Acrobot: -353.1 +- 120.3 (compared to -500 for PPO and -245.4 +- 48.4 for XLVIN with hand-crafted graphs)
>     - MountainCar: -185.58 +- 8.1 (compared to -200 for PPO and -168.87 +- 24.7 for XLVIN with hand-crafted graphs)
>     - Freeway: see the figure here: https://anonymous.4open.science/repository/5dc1ee33-d391-4718-881d-5c3ad77e2b47/Freeway_scores.png (XLVIN-CP is pre-trained with hand-crafted graphs, XLVIN-R with random deterministic graphs).
> - The results indicate that, even when trained on randomly generated graph structures, the XLVIN executor is capable of solving all of the considered environments more rapidly and robustly than the PPO baseline. Hence our method need not be strongly dependent on knowing the specifics of the underlying MDP, at least on the environments evaluated here. In essence, as long as our executor learns a “correct” latent value iteration rule, it should be applicable.
> - We have started work on setting up further ablation experiments. Ideally, and resources permitting, we would attempt replacing the TransE loss with the World Model loss from Ha & Schmidhuber, or our executor model with ATreeC from Farquhar _et al._, (which, being a policy-gradient method, is likely the most meaningful). Additionally, we can report loss values over rollout steps for the various world models.
> - Thank you for the chicken-and-egg remark about world model learning, which we agree to discuss more explicitly in the paper. Whenever appropriate, the TransE model was pre-trained with observations from a random policy, and fine-tuned using the observations collected by the current policy network. That is, for each set of trajectories collected by our XLVIN policy, the PPO loss and TransE loss are optimised jointly. While we found this setup to work well for our purposes, we appreciate how different environments may require a more careful handling of the exploration/exploitation tradeoff.
> We will make all of this discussion, along with our data collection setup, explicit in the manuscript.
>
> Please let us know if you feel satisfied with the direction of our efforts in response to your review. We are very open for further suggestions.

---

> ### Author Response · Authors · 2020-11-23
> **Revision posted, and additional experiments**
>
> Thank you once again for your kind review and constructive comments!
>
> We have now revised the paper on OpenReview, incorporating all of the changes mentioned in our initial reply, and including further experiments, studies and discussions.
>
> Specifically, with respect to your review, we have performed follow-up ablation studies as follows:
> - We found that representations obtained using the pixel-space reconstruction-based World Model (e.g. Ha & Schmidhuber) were not useful for use within the XLVIN. We observed that most of our state encodings in Atari converged to a fixed-point, and that the pixel-space reconstructions completely ignored the foreground observations. This aligns with prior investigations of VAE-style losses on Atari, which found they tend to overly focus on reconstructing the background and were less predictive of RAM state than latent-space models, as well as randomly-initialised CNNs ("Unsupervised State Representation Learning in Atari", Anand _et al._, NeurIPS 2019). This comparison stands in favour of our approach to using a transition model optimised purely in the lower-dimensional latent space. We provide figures in the paper (Appendix H) illustrating the pixel-reconstructions our VAE model obtains.
>
> - Using the ATreeC model of Farquhar _et al._ within our pipeline generally yields results that are less powerful in low-data scenarios. Namely, the model fails to solve the continuous-state environments such as Acrobot and MountainCar, and is only marginally improving on the PPO baseline on CartPole (remaining significantly behind XLVIN). Within the studied Atari environments, it remains consistently behind XLVIN over the first 500,000 transitions for all environments, and stays behind even the PPO baseline on Enduro. We hypothesise that this is due to the fact that ATreeC strongly relies on scalar-based predictions in every node, and in the lower-data regimes, such predictions are not accurate enough to support a strong policy.
>
> Should you have any additional comments at this stage, please let us know!

---

> > ### Comment · AnonReviewer4 · 2020-11-24
> > **Thanks for the Clarifications and the New Results**
> >
> > Thank you for the detailed response. The revisions to the text have greatly improved the clarity of the paper. The comparisons with new baselines are great and have definitely made the submission stronger.
> >
> > I will change my rating to acceptance.

---

> > > ### Author Response · Authors · 2020-11-25
> > > **Thank you!**
> > >
> > > Thank you for your reply!
> > > We are very glad that our updates have addressed your comments appropriately.

---

### Author Response · Authors · 2020-11-23
**Summary of revisions made to the paper in the discussion period**

We have just revised our paper on OpenReview, hoping that we have properly addressed the comments of the reviewers - and that our overall contribution, quality and clarity is now improved! We would like to thank all the anonymous reviewers once again for their thoughtful comments on our paper.

We provide a summary of the changes made to the paper:

* We make explicit that our XLVIN model's expansion policy, as presented, works on discrete action spaces (remark in Section 1), and provide a discussion on its complexity and performance with respect to depth, as well as how to extend the model to large action spaces or continuous actions by leveraging a rollout policy ("Discussion" paragraph in Section 3.2).

* We emphasise our model's novelty further by updating the final bullet point (on neural algorithmic execution) in Section 1, as per our discussion with the reviewers.

* We update Figure 1 to explicitly decouple and sequentialise the tree construction with the neural execution process, hoping to improve its clarity.

* We make explicit the way in which the data is collected for training our transition model, and provide a discussion on the likely exploration/exploitation tradeoff with learning world models in general, in Section 3.3.

* We include and explicitly reference all experiments with random deterministic graphs throughout Section 4 (as XLVIN-R when comparing against the executor pretrained on CartPole-style graphs, which we denote XLVIN-CP), highlighting that we need not generate graphs that are aligned with the underlying MDP structure in order to recover gains from using XLVIN.

* We expand our Atari experiments to include two more challenging environments (in terms of action spaces or rich visual observations): Alien and Enduro, and as per the previous point, pre-training XLVIN executors on random deterministic graphs for both.

* We incorporate two ablation studies in Section 5:
    - First, we incorporate world models relying on pixel-based reconstruction (VAE-style), mirroring prior findings that find them unfavourable in Atari, and providing a qualitative explanation of such in Appendix H.
    - As an additional implicit planning baseline, we perform experiments with the ATreeC-style computation within our pipeline, finding that it is underperforming in low-data environments, and provide a possible discussion on the causes of this (overreliance on scalar predictions in every node, as opposed to latent-space updates employed by XLVIN).

* [Second revision] We provide experiments on an additional discrete grid-world environment (_contextual maze_), where the environment rules of mazes are dictated by global context, which proves difficult for slicing-based VIN models to incorporate. The experiments are detailed in Appendix G and referenced and discussed in Section 4.2.

* Various typos, grammar and style fixes throughout the document.

Should you have additional comments on any of the above, please let us know.

---

### Decision · Program_Chairs · 2021-01-07
**Final Decision**

**Decision:**

Reject

**Comment:**

The work extends the line of work based on value iteration networks. The main goal is to extend VINS to continuous and partially observable state spaces. The approach combines self-supervised contrastive learning and graph representation learning with VINs to address these issues. Reviewers liked the premise of the paper and had several follow-up clarifications.  The authors provided the rebuttal and addressed some of the concerns. However, upon post rebuttal discussion, the reviewers decided to maintain their score. While everyone recommended weak acceptance, no one championed the paper. This was primarily due to the concerns in the empirical analysis. It is not clear that XLVINs are clearly outperforming VINs and Graph-VINs in all settings. All baselines are not present in all the environments, so it is difficult to draw a consistent conclusion. The paper is in a good state but not fully polished to infer clear conclusions about the effectiveness of the proposed approach. Please refer to the feedback below for more details. We believe strengthening the experimental results section will turn this paper into a very strong submission.